# Cotton Fabric Modified with a PAMAM Dendrimer with Encapsulated Copper Nanoparticles: Antimicrobial Activity

**DOI:** 10.3390/ma14247832

**Published:** 2021-12-17

**Authors:** Desislava Staneva, Daniela Atanasova, Ani Nenova, Evgenia Vasileva-Tonkova, Ivo Grabchev

**Affiliations:** 1Department of Textile, Leather and Fuels, University of Chemical Technology and Metallurgy, 1756 Sofia, Bulgaria; grabcheva@mail.bg (D.S.); d.atanasova1@abv.bg (D.A.); ani.nenowa@gmail.com (A.N.); 2The Stephan Angeloff Institute of Microbiology, Bulgarian Academy of Sciences, 1113 Sofia, Bulgaria; evaston@yahoo.com; 3Faculty of Medicine, Sofia University, St. Kliment Ohridski, 1407 Sofia, Bulgaria

**Keywords:** textile, metallodendrimer, nanoparticles, antimicrobial activity, photodynamic therapy

## Abstract

A new methodology for modifying textile materials with dendrimers containing nanoparticles was developed. This involved a combination of eosin Y and *N*-methyldiethanolamine (MDEA) for reducing the copper ions in the dendrimer complex by enabling a photochemical reaction under visible light and ambient conditions. The conversion of copper ions into nanoparticles was monitored using scanning electron microscopy (SEM) and by performing colorimetric, fluorescence, and electron paramagnetic resonance (EPR) studies. Regardless of the concentration of the photoinitiator eosin Y, it discolored completely upon illumination. Three types of cotton fabrics were compared as antimicrobial materials against *Bacillus cereus*. One of the fabrics was dyed with a first-generation PAMAM dendrimer which had been functionalized with eight 1,8-naphthalimide fluorophores. Another fabric was dyed with a dendrimer–copper complex, and the third was treated by conversion of the complex into copper nanoparticles encapsulated into the dendrimer. An enhancement in the antimicrobial activity of the textiles was achieved at higher dendrimer concentrations, under illumination with visible light. The fabric modified with the copper nanoparticles encapsulated inside the dendrimer exhibited the best antibacterial activity because it had two photosensitizers (PS), as both 1,8-naphthalimide fluorophores and copper nanoparticles were contained in the dendrimer molecules. The presence of oxygen and suitable illumination activated the photosensitizers to generate the reactive oxygen species (singlet oxygen (^1^O_2_) and other oxygenated products, e.g., anion radicals, hydroxyl radicals, and hydrogen peroxide) responsible for destroying the bacteria.

## 1. Introduction

There is a long history of treating materials with antimicrobial agents. However, nowadays, resistance of microorganisms to many widely used products is observed more and more frequently; therefore, research on new compounds and methodologies to combat this has become crucial for quality of life [1]. This is of particular importance as far as textile materials are concerned, since they have a very specific surface area and, if hydrophilic, retain moisture, hence becoming an ideal substrate for microorganisms [2]. There are a variety of methods for antimicrobial treatment of textiles so that the resulting materials can find various safe applications, e.g., as wound dressings, hospital textiles, bed linen, towels, filters, and other medical supplies [3]. Some of the compounds intensively studied and applied for the purpose are metal complexes of various organic ligands [4]. Copper ions are among the most commonly used in their design and synthesis, as they are found in living organisms as biologically important metal ions [5,6,7]. In recent years, dendrimer molecules have been successfully used as multidentate ligands, due to their specific structure [4,8]. Dendrimer complexes obtained previously with different ions (Ag(I), Pt(II), Pd(II), Cu(II), or Zn(II)) have demonstrated great potential as novel highly effective agents with antimicrobial activity [9,10,11,12]. These compounds have also been studied as promoters of metal ion reduction in the dendritic complex with a view to encapsulation of the metal nanoparticles [13,14]. In most cases, sodium borohydride was used as a reducing agent. It has been found that the biological activity of Ag, Cu, Zn, Ni, and Co nanoparticles and their oxides depends on various factors such as size, shape, and the presence of defects in their structure, together with their distribution and interaction with the material. A dendrimer can serve as a template for metal nanoparticles providing control over their size, morphology, and stability. Metal ions react with the functional groups of the dendrimer and, having been reduced, form almost monodisperse nanoparticles smaller than 3 nm. Increasing the particles’ surface-to-volume ratio influences the number of reactive sites on the nanoparticle surface and controls the generated amount of reactive oxygen species (ROS), determining their application as catalysts and antimicrobial agents. Recently, the syntheses of nanoparticles stabilized by either triazole-containing or triazole-free dendrimers have been described [15]. The syntheses were conducted using water-soluble salts of the metal ions and a dendrimer dissolved in deionized water at a standard 1:1 molar ratio of metal ion/triazole ligand. Then, NaBH_4_ was added as a reductant. These materials were prepared by sequestering metal ions within dendrimers, followed by chemical reduction to yield the corresponding zero valency metal nanoparticles, whose size depends on the number of metal ions loaded initially into the dendrimer. The capping of dendrimers on silver nanoparticles and other metal nanoparticles or their composites has further increased their effectiveness due to additive or synergistic actions [16]. Low-temperature conditions and the use of a mild reducing agent favor a kinetics-driven process, and thus the formation of anisotropic nanoparticles [17].

Our previous studies in this area have shown that the ligands themselves (modified poly(amidoamine) (PAMAM) dendrimers with 1,8-naphthalimides) exhibit good antimicrobial properties that improve upon complexing with Zn(II) and Cu(II) ions [18,19,20,21,22,23]. It is of interest to control this biological activity when processing the textile materials by obtaining nanoparticles of suitable structure and reactivity. Encapsulation of copper nanoparticles with a high surface-to-volume ratio by fluorescent dendrimers allows the production of biological agents with high chemical reactivity and is able to improve the antimicrobial properties of the loaded textile material.

This study aims at synthesizing a copper complex of first-generation PAMAM dendrimer functionalized with eight 1,8-naphthalimide fragments. The modification of cotton fabrics with the dendrimer ligand and the metallodendrimer is also described. A combination of eosin Y and MDEA was used for the in situ synthesis of the dendrimer with encapsulated copper nanoparticles on the textile surface. Various factors that influence the antimicrobial activity of the obtained textiles were compared, i.e., the structure and concentration of the antimicrobial agent and the effect that visible light illumination has upon the materials.

## 2. Materials and Methods

### 2.1. Materials

A 100% cotton fabric with a plain weave structure and a weight of 140 g/m^2^ was used throughout the work. The fabric was scoured and bleached and had a warp and weft density of 29 threads/cm and 24 threads/cm. Eosin Y and MDEA were used as obtained from Sigma-Aldrich (St. Louis, MO, USA). The synthesis of PAMAM dendrimer (D) and its spectral characterization were described earlier [24].

### 2.2. Synthesis of [Cu_8_(D)(NO_3_)_16_] Complex

A total of 0.3650 g (0.1 mM) of dendrimer D was dissolved in 50 mL of ethanol, and Cu(NO_3_)_2_.3H_2_O (0.246 g, 1.0 mM) was added to the solution at 25 °C. After 30 min the precipitate was filtered off, washed with ethanol, and dried. The yield was 0.89% (0.450 g).

FTIR was performed at 1640, 1577, 1523, 1426, 1380, 1338, 1229, 1183, 1056, 834, 779, 757, 580, 501 and 462 cm^−1^. Elemental analysis was as follows: C_190_H_240_N_58_O_76_Cu_8_ (5056.9): calculated: C—45.09, H—4.74, N—13.76; found: C—45.18, H—4.63, N—13.62.

### 2.3. Cotton Fabric Functionalization with Fluorescent PAMAM Dendrimer and Its Copper Complex

Polyamidoamine PAMAM dendrimer from the first generation was modified with 1,8-naphthalimide fluorophore (Figure 1) [24] and its copper complex [Cu_8_(D)(NO_3_)_16_] were used for treating the cotton fabric.

The cotton samples were dyed with the dendrimer and its complex containing 8 copper ions [Cu_8_(D)(NO_3_)_1__6_]. The experiments were run in a *N*,*N*-dimethylformamide solution at two different concentrations (0.5% and 1% by weight of the fabric) and at a liquor-to-goods ratio of 1:3. The cotton fabrics were impregnated with the solutions obtained and dried at 60 °C for 20 min, then washed with water. Next, the four obtained textile samples (D1, D2, C1, and C2) were impregnated with an aqueous solution of a reducing agent (eosin Y and MDEA). As the concentrations of the dendrimer and its complex were different, the concentration of reductant was also altered (see Table 1). Afterwards, each of the four newly impregnated samples was cut into two. One of each group was illuminated with visible light, at room temperature, in air. These four obtained textile samples were named DR1I, DR2I, CR1I, and CR2I, respectively. The other samples, DR1N, DR2N, CR1N, and CR2N were stored in the dark prior to performing the colorimetric analysis. The studied dendrimer and metallodendrimer had high molecular weights (3560.3 Da for dendrimer D and 5056.9 Da for the metallodendrimer), and both were insoluble in water. This determines their stability after deposition and fixation on the surface of cotton fabrics. In aqueous solution, no migration of dendrimers was observed.

### 2.4. Colorimetric Analysis

The color characteristics (CIE (L*, a* and b*) color space and color depth (K/S)) of the fabrics dyed with the dendrimer and its copper complexe were investigated and compared to their spectral characteristics after coating with a solution of eosin Y and MDEA, as well as after illumination with visible light. A comparison of the results achieved when altering the concentration of the dendrimers and the results with the reducing agent was made.

The intensity of the color was expressed on the basis of the Kubelka–Munk equation [25,26]:(1)$$\frac{K}{S}=\frac{{\left(1-R\right)}^{2}}{2R}$$
where *K* is the light absorption, *S* is the light scattering, and *R* is the reflectance, expressed in fractional form. The *K*/*S* vs. wavelength curve was used for the color characterization of the textile samples. The color measurements were carried out on a Datacolor Spectraflash SF300 apparatus.

### 2.5. Material Characterization

The fluorescence spectra were acquired using a Cary Eclipse fluorescence spectrophotometer at a resolution of 2 nm. Permeation experiments were performed by placing the samples in the diagonal direction in an empty fluorescence cuvette. The excitation and fluorescence spectra of cotton fabrics dyed with PAMAM dendrimer and its copper complex, and those of the same fabrics after treatment with eosin Y and MDEA, were recorded. An IRAffinity-1 spectrophotometer (Shimadzu Co., Kyoto, Japan) was used for monitoring the IR spectra. A Jeol JSM-5510 scanning electron microscope was used for SEM characterization of the samples. The EPR spectra of the powdered dendrimer and its copper complex, and the textile samples D2, C2, and CR2I, were recorded as the first derivative of the absorption signal of a Bruker EMXplus EPR spectrometer in the X-band (9.4 GHz), within the temperature range of 120–295 K.

### 2.6. Antimicrobial Activity of Fabrics

The antimicrobial activity of the fabrics (D1, C1, CR1I, D2, C2, and CR2I) was investigated against Gram-positive *Bacillus cereus* in meat-peptone broth. Test tubes with 2.0 mL of sterile meat-peptone broth were inoculated overnight with bacterial culture and incubated for 2 h at 28 °C under shaking. Then, the textile samples (10 mm × 10 mm) were inserted into the test tubes. After 24 h of incubation in the dark at 28 °C under shaking (240 rpm), the samples were removed, and the bacterial growth was determined by measuring the turbidity of the medium at 600 nm (OD_600_). Similar studies were conducted under illumination with a lamp (HL 8325, 25 W, 1230 lumen, 6400 K).

## 3. Results

### 3.1. IR Characterization

The IR spectra of dendrimer D and its copper complex are plotted in Figure 1. Their comparison reveals that the characteristic bands have approximately the same values. The main difference is in the 1380–1260 cm^−1^ area, where the bands for the C-N bonds (1345 cm^−1^) occur. The spectrum of D in this region has an overlap of the signals from these bands with the signals from the nitrate groups (-NO_3_) which leads to the expansion of the resulting band and the shifting of the maximum to lower frequencies (1330 cm^−1^). This proves the presence of Cu-NO_3_ in the structure of the complex. As can be seen from the structure of dendrimer D, copper ions can form a coordinate bond with different groups, both from the core and periphery of the dendrimer. In this case, probably four copper ions form complexes in the dendrimer periphery, as shown in Figure 1. The copper ion coordinates with two nitrogen atoms of the *N*,*N*-dimethylamino groups of the substituent at the C-4 position of the 1,8-naphthalimide units, and with two of the imide carboxyl groups. As a result, the polarization changes, and the intensity of the characteristic band for C=O at 1640 cm^−1^ decreases. The low-intensity signal from the vibration of the N-CH_3_ bond at 1435 cm^−1^ also weakens. Four copper ions were coordinated inside the dendrimer molecule. In this case, the coordination takes place with the amide oxygen and nitrogen atoms of the dendrimer structure. As a result, the intensity of the band at 1525 cm^−1^ becomes lower, compared to that of the dendrimer ligand. A similar dependence was observed for other metallodendrimers studied at our laboratory [18].

### 3.2. Scanning Electron Microscope Analysis of Dendrimer D and [Cu_8_(D)(NO_3_)_16_]

Scanning electron microscopy was used to study the microstructure of dendrimer ligand D and the copper complex [Cu_8_(D)(NO_3_)_16_] in the solid state. Figure 2A shows that dendrimer D has a layered structure with irregular shapes. It consists of irregularly shaped monodimensional flakes at the nanometer scale, which tend to aggregate into a layered structure. The formation of a copper complex alters the dendrimeric morphology, as shown in Figure 2B. The resulting powder has an amorphous structure and a significantly reduced density. It is formed by nano-sized grains with an almost spherical shape, which are better seen at a magnification of 60,000 times (Figure 2C). Probably, the formation of a copper complex changes the conformation of the dendrimer. This affects the molecular interactions and is the reason for the reduced density of the solid sample. A similar dependence was determined in the case of the copper complex of a 1,8-naphthalimide-modified PAMAM dendrimer of generation zero [18].

### 3.3. EPR Characterization of the Dendrimer, Its Cu(II) Complex and Textile Samples Containing the Dendrimer, the Copper Complex, and the Dendrimer with Encapsulated Copper Nanoparticles

The spectra in Figure 3 show that a signal resembling the signal in the spectrum of the powdered dendrimer is observed in the EPR spectrum of sample D1 (cotton fabric dyed with dendrimer ligand D). The EPR spectrum of sample CR1I, containing copper nanoparticles encapsulated by the dendrimer structures, does not show distinctive features compared to sample D1, and this is clearly visible from the overlaid spectra in the inset in Figure 3. In a comparative study of the third spectrum, obtained from sample C1 which contains copper–dendrimer complex, an additional signal can be identified within the magnetic field range of 3174 to 3308 Gauss, g ≈ 2.075, appearing parallel to the previously reported signal for the dendrimer [18,21]. While this signal is completely missing in the spectra of cotton fabrics D1 and CR1I, the signal is clearly visible in the spectrum of the dendrimer complex in its powdered form. This signal is evidence of the presence of Cu^2+^ ions in the complex and in the textile material.

This allows the conclusion that no copper ions are present in sample CR1I because they were successfully reduced to nanoparticles. Unfortunately, the interfering effect of the pure dendritic ligand signal does not allow simulation and observation of the EPR characteristics of the spectrum of the copper ion in the complex immobilized on the cotton fabric. A narrow signal (ΔHpp ≈ 1.6 mT) in the range of g-factor values characterizing the free radicals, i.e., g ≈ 2.006, was also found in the spectra of the textile samples.

Figure 4 shows the EPR spectra of the dendrimer, its complex, and the cotton fabrics D1, C1, and CR1I at 120 K. The comparative analysis of the three fabric spectra does not show any differences between them. The dominant signal at this low temperature is the one originating from the dendrimer ligand.

### 3.4. Scanning Electron Microscope Analysis of Cotton Fabric Treated with Dendrimer D and [Cu_8_(D)(NO_3_)_10_]

Figure 5 shows the SEM micrographs of the pure cotton fabric (Figure 5A) and the fabric after its treatment with dendrimer D (Figure 5B), metallodendrimer [Cu_8_(D)(NO_3_)_16_] (Figure 5C), and dendrimer with encapsulated copper nanoparticles (Figure 5D). As can be seen, the neat cotton fibres have a surface roughness on a micro scale resulting from the presence of microfibrils [27]. This morphology remains the same upon loading the dendrimers, as they are deposited over the fiber surfaces. As well as their even distribution, one can see aggregates of different sizes. This trend is also maintained in the copper complex, but the aggregation processes are less pronounced (Figure 5C). The reduction of Cu(II) ions included within the dendrimer molecule yields copper nanoparticles, which are evenly distributed over the cotton fabric surface (Figure 5D).

### 3.5. Comparative Colorimetric Study of the Cotton Fabric Dyed with Fluorescent Dendrimer, Its Copper Complex and after Impregnation with Aqueous Solution of the Reducing Agents (before and after Illumination with Visible Light)

In our previous studies on the synthesis of silver nanoparticles, we used modified eosin Y (photoinitiator) and *N*-methyldiethanolamine (co-initiator) for radical photopolymerization [28]. In the present study, the system (eosin Y and MDEA) was exploited as a reducing agent for the copper ions involved in the dendrimer complex. The light-induced reaction changes the chromophore system of eosin Y, so that it discolors. These processes were followed using colorimetric analysis. The K/S vs. wavelength curves for the textile samples treated with the dendrimer or its complex are presented in Figure 6. As can be seen, the cotton fabrics dyed with the dendrimer absorb light with a maximum at 440 nm (samples D1 and D2). The impregnation with an aqueous solution of the reducing agents leads to the appearance of a new band with a maximum at 520–530 nm, which is characteristic of eosin Y. Since the amount of eosin Y is greater in sample DR2N, the intensity of the absorption is higher than in sample DR1N; hence, the band at 440 nm is bathochromically shifted toward 450 nm. After illumination with visible light, the bands for eosin Y disappear. The band at 450 nm is hypsochromically shifted toward 440 nm for sample DR2I, and the absorption intensity for both samples DR1I and DR2I is weakened.

The spectra of the cotton fabrics dyed with the dendrimer complex are approximately the same with small but intrinsic differences. The absorption maximum for the samples dyed with the dendrimer complex is also at 440 nm (C1 and C2). The band for eosin Y in the spectra of CR1N and CR2N appears at the same wavelength as in the spectra of DR1N and DR1N. However, after illumination, the intensity of absorption at 440 nm increases for both samples (CR1I and CR2I), which can be explained by the formation of copper nanoparticles.

The data for the color space coordinates (light–dark (L*), red−green (a*) and yellow−blue (b*) values) of the samples were compared and are shown in Table 2. The a* and b* values of fabrics dyed with dendrimer D and its copper complex are very close to one another. L* is lower for fabrics C1 and C2 than for D1 and D2, respectively. Dyeing with a higher dendrimer concentration or a dendrimer complex gives the fabrics darker, redder, and yellower colors. After treatment with the reducing system (eosin Y and MDEA) and illumination, the colors of all samples are less yellow, and b* decreases while L* and a* values increase. More significant changes are observed between samples DR2I and CR2I than between DR1I and CR1I.

### 3.6. Comparative Fluorescence Study of Cotton Fabric Dyed with Fluorescent Dendrimer, Its Copper Complex and after Its Impregnation with Aqueous Solution of Reducing Agents

Textile samples D1 and C1, and DR1I and CR1I, were subjected to fluorescence analysis after illumination with visible light (Figure 7). The cotton fabric dyed with the dendrimer modified with 1,8-naphthalimide fluorophore was excited at 450 nm and emitted light at λ_max_ = 525 nm. As a result of the complexation of copper ions with the dendrimer, the fluorescence intensity of fabric C1 was quenched drastically. After treatment with a solution of eosin Y and MDEA and upon subsequent illumination with visible light, the fabric dyed with the dendrimer (DR1I) had lower fluorescence intensity compared to that of sample D1. This phenomenon is due to the alkaline action of MDEA and to the occurrence of photoelectronic transfer. The procedures undergone by fabric CR1I dyed with the dendrimer complex, i.e., eosin Y and MDEA treatment as well as light illumination, enhanced its fluorescence intensity compared to the intensity measured for the sample with dendrimer (D1). This fact can be explained by the restoration of fluorescence emission after the reduction of copper ions, leading to the formation of nanoparticles.

### 3.7. Photo-Oxidation of Potassium Iodide

The iodometric method was used for the detection of singlet oxygen (^1^O_2_) during the illumination of the dyed cotton fabrics D2, C2, and CR2I. In this case, the modified 1,8-naphtalimide dendrimers acted as a photosensitizer (PS) and participated in the generation of singlet oxygen (^1^O_2_), which reacts with colorless I^−^ to give yellow-colored I^3-^, according to the following mechanism [29,30,31]:
Do + hν →^1^Dlight excitation^1^D →^3^Dinter system crossing^3^D + ^3^O_2_ → Do + ^1^O_2_energy transfer^1^O_2_ + I^−^ → IOO^−^ → IOOHIOOH + I^−^ → HOOI_2_^−^ → I_2_ +HO_2_^−^ → I_3_^−^ + H_2_O_2_ + OH^−^
where D is the dendrimer used as a PS in the ground state (Do) and in the singlet (^1^D) and triplet excited state (^3^D).

The samples of cotton fabric (1 cm/1 cm) in 5 mL aqueous KI (0.5 M) solution were treated with a Newport solar simulator with the following characteristics: 185–1100 nm, Xe lamp, 150 W, at 25 cm sample distance. The absorption spectra in the 270–500 nm range were recorded every 5 min.

The dependence of the absorption as a function of the illumination time of the solution in which the cotton fabric D1 was placed, is shown in Figure 8. As the figure shows, the spectral profile of the solution is the same and only the absorption is different. On the other hand, it is similar to that in which the PS is in solution [31]. This means that the dendrimer generates singlet oxygen in a solid form and not only in solution. Before illumination, the KI solution showed no absorption in the range λ_A_ = 270–550 nm. After illumination of the solution, two well-defined peaks typical of the I^3-^ formed in the solution were observed in the absorption spectrum: one at 288 nm and the other at 352 nm [31]. As can be seen from Figure 8, the absorption at these maxima increased with the time of illumination, which is an indicator that the concentration of (^1^O_2_) also increased. Similar dependencies were found using the other cotton fabrics tested. The dependence of the absorption at λ_A_ = 352 nm as a function of the illumination time for the treated cotton fabrics D2, C2, and CR2I, is shown in Figure 9.

### 3.8. Antimicrobial Activity of Modified Cotton Fabrics

Several factors are very important for achieving a good antimicrobial activity for a given material, namely: surface modification of the cotton fibers, distribution and agglomeration of the antimicrobial dendrimer substance on the fiber surfaces, and changing the hydrophilic/hydrophobic properties. The uniform distribution of dendrimers on the surface of the cotton fibers leads to inhibition of biofilm formation [19,23]. In addition, dendrimers interact with microbial cells and affect their adhesion (which is the first stage of biofilm formation) and also prevent the formation of an extracellular matrix [32].

The antimicrobial activity of the unmodified cotton fabric was used as a control and compared to the modified fabrics (D1, C1, CR1I and D2, C2, CR2I) against the Gram-positive bacterium *Bacillus cereus*. The influence of light on the antimicrobial activity of textile materials was also studied. The results are presented in Figure 9. As can be seen, the antimicrobial activity of all the treated fabrics was much better upon light illumination than after being kept in the dark. Unlike in our other studies, the fabrics treated with the copper–dendrimer complexes (C1 and C2) exhibited a slightly weaker antimicrobial activity than those treated with the dendrimer ligands (D1 and D2) [18,19,28]. However, this activity increased after the reduction of the copper ions. The best result, showing complete inhibition of bacterial growth, was obtained for sample CR2I containing nanoparticles upon light illumination. This result was much better than the one obtained for the same cotton fabric kept in the dark. In this case, the antimicrobial activity of the CR2I fabric was due to the synergetic effect of two factors. On the one hand, the 1,8-naphthalimide fluorophores from the dendrimer molecule exhibit biological activity, and on the other hand, the nanoparticles also have an effect. In both cases, during the irradiation with light, ROS such as OH·, O_2_·, and H_2_O_2_, are generated. These species are very active in killing bacteria without harming non-bacterial cells [33]. Similar results were obtained using photoactive polypropylene imine dendrimers modified with 3-bromo-4-dimethylamino-1,8-naphthalimide after deposition on a cotton cloth and illumination with visible light [34].

## 4. Conclusions

A new methodology for modifying textile materials with dendrimers containing copper nanoparticles was developed. New agents (eosin Y and MDEA) were successfully applied for reducing the copper ions in the dendrimer complex by carrying out a photochemical reaction under the conditions of visible light, room temperature, and air. Regardless of the concentration of the photoinitiator eosin Y, it completely degraded, as a result of a change in its chromophore system, during illumination. The process of reduction of copper ions to nanoparticles was followed using colorimetric, fluorescence and EPR investigations. The antibacterial activity of cotton fabrics modified with dendrimers was tested against *B. cereus* in the dark and after illumination with visible light. The results demonstrated a strong increase in the antimicrobial activity of nanoparticle-containing fabrics that might be due to an increase in the concentration of the reactive oxygen species responsible for destroying the bacteria. The generation of such reactive species by illumination of cotton fabrics with visible light was proven using the iodometric method.

## Data Availability

Not applicable.

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
