# Peer review of "Cotton Fabric Modified with a PAMAM Dendrimer with Encapsulated Copper Nanoparticles: Antimicrobial Activity"

_materials, 2021, doi:10.3390/ma14247832_

Round 1

Reviewer 1 Report

The work describes a new method to antimicrobial finishing  apply on to cotton textiles. The methods are clearly described. Sentences are sometimes very long and written, which makes the work somewhat hard to read.

However, it has scientific relevance and is within the scope of this journal. So that the publication can take place, I advise some corrections and improvements.

Some general points about the work that should be reviewed for publication

Abstract
Check the Abstract grammar  and emphasize at the end the objective achieved with this work, especially regarding the antimicrobial effect achieved.

Materials
Regarding the description of the textile article (fabric? knitwear? nonwoven? Add type of weave, yarn title, density and textile structure...)

Methodology
The methodology is confused, there is a need to clarify which equipment and procedures were used to obtain the FTIR and the SEM.

Furthermore, in the case of colorimetry, CIE lab data can be used as results and used to increase the discussion regarding colorimetric concepts.

Another important point is the insertion of the term “White” (standard) in the Kubelka-Munk equation, since the background of the textile article must be evaluated in order to have the result only of the finishing and not of the coloring of the textile.

Regarding the antimicrobial evaluation method, there are no explanations as to why this microorganism was chosen.

Results
The characterization of the textile article must be better presented. Photos of the sample before and after the process need to be inserted. As only the SEM is not possible to verify the overlap of the textile finish. These photos will help with understanding and explanation, as well as the CIE Lab discussion.

Another characterization to be performed to prove the deposition would be the FTIR-ATR of the textile article, showing the surface has the finish and not just impurities.

In textile, one of the main characteristics to be evaluated for textile finishing is its durability against washing and friction. Thus, it would be interesting to evaluate the finish after exposure of the textile to these two factors, separately.

Author Response

Check the Abstract grammar and emphasize at the end the objective achieved with this work, especially regarding the antimicrobial effect achieved.

The abstract has been rewritten. 

Materials

Regarding the description of the textile article (fabric? knitwear? nonwoven? Add type of weave, yarn title, density and textile structure...)

The fabric of 100% cotton with plain weave structure and weight of 140 g/m2 was used throughout the work. The fabric is scoured and bleached and have the warp density 29 threads/cm and 24 threads/cm.

Methodology

The methodology is confused, there is a need to clarify which equipment and procedures were used to obtain the FTIR and the SEM.

The equipment used are: IRAffinity-1 spectrophotometer (Shimadzu Co., Kyoto, Japan) has been used for monitoring of the IR spectra.

Jeol JSM-5510 scanning electron microscope has been used for SEM characterization of sample and they have been added to the text.

Furthermore, in the case of colorimetry, CIE lab data can be used as results and used to increase the discussion regarding colorimetric concepts.

The CIE lab data have been added to the text

Another important point is the insertion of the term “White” (standard) in the Kubelka-Munk equation, since the background of the textile article must be evaluated in order to have the result only of the finishing and not of the coloring of the textile.

The K/S  for the initial cotton fabrics was added.

Regarding the antimicrobial evaluation method, there are no explanations as to why this microorganism was chosen.

Bacillus cereus was adopted as a model Gram-positive bacterium for laboratory studies.

Results

The characterization of the textile article must be better presented. Photos of the sample before and after the process need to be inserted. As only the SEM is not possible to verify the overlap of the textile finish. These photos will help with understanding and explanation, as well as the CIE Lab discussion.

The CIE Lab characteristics has been discussed.

Another characterization to be performed to prove the deposition would be the FTIR-ATR of the textile article, showing the surface has the finish and not just impurities.

Due to the small amount of dendrimers on the surface of the cotton fabric, the sensitivity of the IR spectrometer does not allow us to see the difference between the treated and untreated material.

In textile, one of the main characteristics to be evaluated for textile finishing is its durability against washing and friction. Thus, it would be interesting to evaluate the finish after exposure of the textile to these two factors, separately.

The studied dendrimers and metallodendrimers are with high molecular weights and they are insoluble in water.  This determines their stability after the deposition and fixation on the surface of cotton fabrics.  In aqueous solution any migration of dendrimers has not been observed.

Reviewer 2 Report

The manuscript describes, “Cotton fabric modified with a PAMAM dendrimer 2 encapsulated copper nanoparticles: Antimicrobial activity” which is suitable for Journal of Materials. Anyhow, the reviewer would like to make the following comments

1- Page 2, Line 47, it is recommended that change the “form” word to morphology

2-why did the efficiency nanoparticles increase with increasing the surface-volume ratio enhances?

3- Justify why the density of the sample has been reduced significantly in Fig.2b?

4-Line 330, cite the suitable references after “Unlike our other studies”

5- Justify why, the light irradiation improved the antimicrobial activity

Author Response

1- Page 2, Line 47, it is recommended that change the “form” word to morphology

Done

2-why did the efficiency nanoparticles increase with increasing the surface-volume ratio enhances?

Done

3- Justify why the density of the sample has been reduced significantly in Fig.2b?

Probably the formation of a copper complex changes the conformation of the dendrimer. This affects the molecular interactions and is the reason for the reduced density of the solid sample.

4-Line 330, cite the suitable references after “Unlike our other studies”

Done

5- Justify why, the light irradiation improved the antimicrobial activity.

In this case, reactive oxygen species (ROS) are generated, which kill bacteria more efficiently. Explanation is given in the text.

Reviewer 3 Report

This work reported a methodology for modifying textiles material with dendrimers containing nanoparticles. The antimicrobial activity of the loaded textile samples has been improved by enhancing the concentration of the dendrimers and under irradiation with visible light. This manuscript is well organized and the conclusions are supported by the solid data. There are still some issues that need to be addressed before publication. The detailed comments are listed below.

  • How does the concentration of the Dendrimer affect the morphology of nanoparticles? Please supplement the information.
  • What is the molecular weight of the Dendrimer?
  • The structure of the dendrimer D needs to be further confirmed by other measurements. Only FTIR is not enough.
  • What about the anti-washing of the coatings on the surface of cotton fabric?
  • To highlight the process, the antimicrobial activity of the coating needs to be compared with the reported results.

Author Response

How does the concentration of the Dendrimer affect the morphology of nanoparticles? Please supplement the information.

It is not investigated.

What is the molecular weight of the Dendrimer?

The molecular weight of dendrimers is (3560.3 Da for dendrimer D and 5056.9 Da for metallodendrimer)  

The structure of the dendrimer D needs to be further confirmed by other measurements. Only FTIR is not enough.

The synthesis and characterization of dendrimer D has been reported in a previous publication (see ref. 24). In the present study, we report the synthesis of its copper complex. Due to the paramagnetic nature of copper, NMR spectroscopy can not be used, so we used the most reliable analytical method for copper complexes, such as EPR.

What about the anti-washing of the coatings on the surface of cotton fabric?

The studied dendrimers and metallodendrimers are with high molecular weights and they are insoluble in water.  This determines their stability after the deposition and fixation on the surface of cotton fabrics.  In aqueous solution any migration of dendrimers has not been observed.

To highlight the process, the antimicrobial activity of the coating needs to be compared with the reported results.

Done in the text

Round 2

Reviewer 3 Report

I recommend the acceptance of the revised manuscript for publication.